# Hydrological Effects of Agricultural Water Supplies on Paddy Fields using Surface–Groundwater Integrated Model

**Sanghyun Park [1], Hyeonjun Kim [1,2,*], Choelhee Jang [2] and Deokhwan Kim [2]**

1 Department of Civil and Environmental Engineering, University of Science & Technology, 217, Gajeong-ro, Yuseong-gu, Daejeon 34113, Korea; sanghyun0385@kict.re.kr

2 Korea Institute of Civil Engineering & Building Technology, 283, Goyang-daero, Ilsanseo-gu, Goyang-si 10223, Korea; chjang@kict.re.kr (C.J.); kimdeokhwan@kict.re.kr (D.K.)

* Correspondence: hjkim@kict.re.kr; Tel.: +82-31-910-0003

**Abstract:** Agricultural water demands are mainly dependent on the supply from groundwater withdrawals and the supply from agricultural reservoirs. To understand the water cycle of the agricultural catchment, it is necessary to consider the actual situation of the water cycle of paddy fields in catchments through accurate hydrological modeling. In this study, streamflow simulations were implemented in consideration of the levee height of paddy fields and the irrigation period for one sub-catchment of the Boryeong Dam catchment using the integrated surface–groundwater model, CAT (Catchment Hydrologic Cycle Assessment Tool). To consider the agricultural reservoirs in modeling, the catchment was divided into the reservoir sub-catchments, upstream sub-catchments, downstream sub-catchments, and irrigated districts of each sub-catchment. This study aims to analyze the hydrological effects of agricultural reservoirs and groundwater pumping on the hydrological cycle of the catchment and on the soil moisture and groundwater level. As a result of the simulations, we found that the direct flow, baseflow, and groundwater recharge of the catchment increased with the agricultural reservoir supply water. In addition, the effect of drought on soil moisture content and groundwater level in the irrigated paddy fields from agricultural reservoirs was evaluated. The soil moisture increased by about 10% according to the water supply of agricultural reservoirs. The groundwater level rapidly decreased due to the groundwater abstraction during the irrigation period; however, it was analyzed that the water supply from agricultural reservoirs is significantly effective in preventing the decrease in the groundwater level in the irrigation season.

**Keywords:** hydrological responses; irrigation; paddy fields; groundwater; soil moisture; CAT model

## 1. Introduction

Rice paddies in monsoon Asian regions account for 87% of the global paddy rice harvested area and 90% of rice production [1]. Since rice is a staple food in Korea, the agricultural industry is active and paddy fields represent more than 8% of the territory [2]. Generally, agricultural catchments consist of paddy fields, where crops are cultivated and produced, as well as forest areas and residential areas, and nearby water supply facilities artificially supply the water necessary for crop growth [3]. The sources of water supply for agricultural activities can be divided into agricultural reservoirs and groundwater pumping. Water shortages due to the excessive pumping of groundwater in rural areas cause water problems for domestic consumption and can also cause a shortage of water needed for agricultural activities, affecting the amount of streamflow in a catchment [4]. In addition, agricultural water supply is an essential resource for agricultural water consumption, accounting for more than 50% of the total water resource consumption, and it is highly dependent on the water supply of agricultural reservoirs [5]. There are currently 17,629 reservoirs nationwide, of which 17,516 are agricultural purpose reservoirs, accounting for the largest proportion [6]. In addition, as 65% of Korean land is composed

of mountainous areas and the river slopes are steep, runoff occurs in a short period, with a low flow rate during the dry season. As the runoff rate variability is relatively large in Korea [7], it is important to manage water resources through the adequate scale of agricultural reservoirs to meet agricultural water consumption requirements in the catchment. The agricultural reservoir stores the streamflow from the upstream basin and supplies agricultural water to the paddy fields during the irrigation period. A portion of the agricultural water supplied to the paddy fields returns to the stream, and the hydrological cycle in the agricultural catchment works in a complex way with the upstream, the downstream, the irrigation districts, and the reservoirs by the operation of the agricultural reservoirs [3]. Therefore, to understand the hydrological cycle processes of the agricultural catchments, each element should be connected. However, there is a high possibility that they will have difficulties in supplying water during drought because the storage capacity of the agricultural reservoirs is low compared to that of the multi-purpose dams [8]. As the research related to the utilization of water supply is insufficient, the high-accuracy estimation of runoff discharges of reservoir catchment, upstream catchment, and downstream catchment should be preceded for stable water supply and drought response of agricultural catchments. In the agricultural catchments, rice is sensitive to growth periods, the climatic environment, and water content situations. In particular, the soil moisture content is an essential factor in the hydrological modeling of agricultural catchments as crops grow by absorbing soil moisture, the main hydrological component of the agriculture industry [9]. The soil moisture content is the amount of retained water in a soil layer, which could be directly affected by drought. Many drought indices have been developed in the agricultural hydrology field, using variables such as precipitation, evapotranspiration, soil moisture, reservoir, and groundwater level that could directly affect crop growth in drought periods [10]. Representative drought indices indicating agricultural drought include the Reservoir Drought Index, which uses the amount of water stored in agricultural reservoirs; the Soil Moisture Index (SMI), which uses the effective moisture percentage of the soil moisture; and the Integrated Agricultural Drought Index [11]. Bae et al. [12] developed the Agricultural Drought Analysis Model and performed drought analysis on soil moisture and agricultural areas. Shin et al. [13] evaluated daily soil moisture estimation and agricultural drought by linking active- and passive-based soil moisture and a soil moisture data assimilation technique. In this study, the SMI index, often applied for agricultural drought evaluation, and the Standardized Precipitation Index (SPI) and the Standardized Groundwater level Index (SGI), using meteorological data and groundwater level [14], are used to select the drought and normal years within the study period to compare the changes in hydrological components by drought. Research on the hydrological changes of Korean catchments according to agricultural reservoirs has been conducted continuously. Lee et al. [15] analyzed the streamflow changes according to the discharge amount of agricultural reservoirs using the Soil and Water Assessment Tool. Kim et al. [16] evaluated the effect of agricultural reservoirs on streamflow in the Anseong-cheon catchment. Lee et al. [17] evaluated the effect of groundwater consumption and reservoirs on streamflow, and Lee and Noh [18] evaluated the streamflow downstream according to the operation of the agricultural reservoir considering climate change scenarios. In addition, Cho et al. [19] evaluated changes in the river ecological environment and the hydrological environment following the construction of multipurpose dams. However, existing studies on the hydrological changes of catchments according to agricultural reservoirs have been mainly conducted on the streamflow of downstream rivers and ecological changes. Applying the conceptual hydrological model has limitations in considering the infiltration and the streamflow changes by the land use of the catchments. To analyze the water cycle on a catchment scale, hydrologic modeling is required considering streamflow changes in evapotranspiration, direct runoff, and baseflow. In this study, the change in water cycle according to the water supply of the agricultural reservoirs in one sub-catchment of the Boryeong Dam catchment was evaluated, and the changes in the soil moisture content and the groundwater level according to the drought period were analyzed using the integrated

surface–groundwater model, which can consider the actual situation of the paddy fields in Korea. In addition, the sub-catchment where the agricultural reservoirs are located was divided into smaller sub-catchments to analyze the hydrological responses considering the agricultural water supplies. Changes in the soil moisture contents and groundwater elevation in paddy fields according to irrigation and non-irrigation periods were compared and analyzed by considering groundwater withdrawals and agricultural reservoir supplies.

## 2. Materials and Methods

### 2.1. Study Area Description

The Boryeong Dam catchment is located in the West Sea basin of the Geum River watershed in Korea. The catchment area is about 162.84 km$^2$, and the average slope of the catchment is about 29.8%. The catchment has steep and mountainous terrain characteristics, including paddy field areas. Even though this catchment is not a regional-scale catchment, it has complex interactive hydrological cycle processes due to human activities, such as groundwater abstraction, agricultural reservoirs, and water intake supply systems from outside the catchment. Administratively, the Boryeong Dam catchment includes Oesan-myeon of Buyeo-gun and Seongju-myeon and Misan-myeon of Boryeong-si. The Ungcheon Stream flowing through the Boryeong Dam catchment has a separate branch from the Geum River, originating from Oesan-myeon of Buyeo-gun and Seongmyeon of Boryeong-si, respectively; they join together at Misan-myeon of Boryeong-si and flow directly into the west sea of Korea. The total length of the Ungcheon Stream is about 36 km, with 23 km located in the Boryeong Dam catchment. About 8.11% of the total area of the Boryeong Dam catchment comprises paddy fields, about 2.26% is urbanized area, and more than 82% is mountainous area (remaining indicates the area of water surfaces in the catchment) [20]. The average annual precipitation in the catchment is about 1244 mm, and the average monthly temperature during the study period is from −4.5 °C to 27.3 °C [21]. The surface soil texture of the Boryeong Dam catchment is dominated by sandy loam and silty loam, and the subsoil texture is mostly composed of loamy sand. The drainage grade of soil is suitable throughout the catchment [22]. The groundwater is abstracted by public electric pumping stations in the catchment. The amount of water supply of agricultural reservoirs was discharged in the Korean irrigation period between April and September.

The Boryeong Dam, which serves as a major water source in the middle-western region of Korea, recorded its lowest storage volume in 2015 due to the continuous shortage of precipitation since 2014. The water storage volume of the Boryeong Dam in 2015 decreased to 18.87% due to the extreme drought that occurred in the central region of Korea. The drought in the central region continued, and the water storage rate reached 8.29% in 2017, which is the lowest water storage level since the construction of the Boryeong Dam [23,24].

Figure 1a shows the location of the three rainfall gauging stations: one water level station at the outlet of the Boryeong Dam, and two agricultural reservoirs over the catchment [25]. Figure 1b shows the Boryeong Dam catchment divided into three sub-catchments, SC_1, SC_2, and SC_3, according to the previous study [24]. The CAT system for the Boryeong Dam catchment was constructed by dividing the catchment into three sub-catchments and then dividing the paddy field areas of each sub-catchment. The sub-catchments' information, such as the areas, slopes, and impervious area ratio, were calculated as the input data of CAT using the DEM map with 30 × 30 m resolution and of the land use map of the National Geographic Information Institute [26]. Figure 2 shows the Samsan Reservoir and the Hwasung Reservoir, which are agricultural purpose reservoirs located in the upstream of Ungcheon Stream in the Boryeong Dam catchment [25], which are located in the SC_2 sub-catchment according to the catchment division of the previous study [24]. In this study, the SC_2 sub-catchment, in which the agricultural reservoirs are located, is the target area for the streamflow simulation using CAT. The total water storage capacity and the effective water storage capacity of Samsan Reservoir are 546,200 m$^3$ and 543,200 m$^3$, respectively. The total water storage capacity and effective water storage ca-

pacity of the Hwasung Reservoir are 724,400 m$^3$ and 720,900 m$^3$, respectively [5]. Detailed information on the Samsan and Hwasung reservoirs is shown in Table 1.

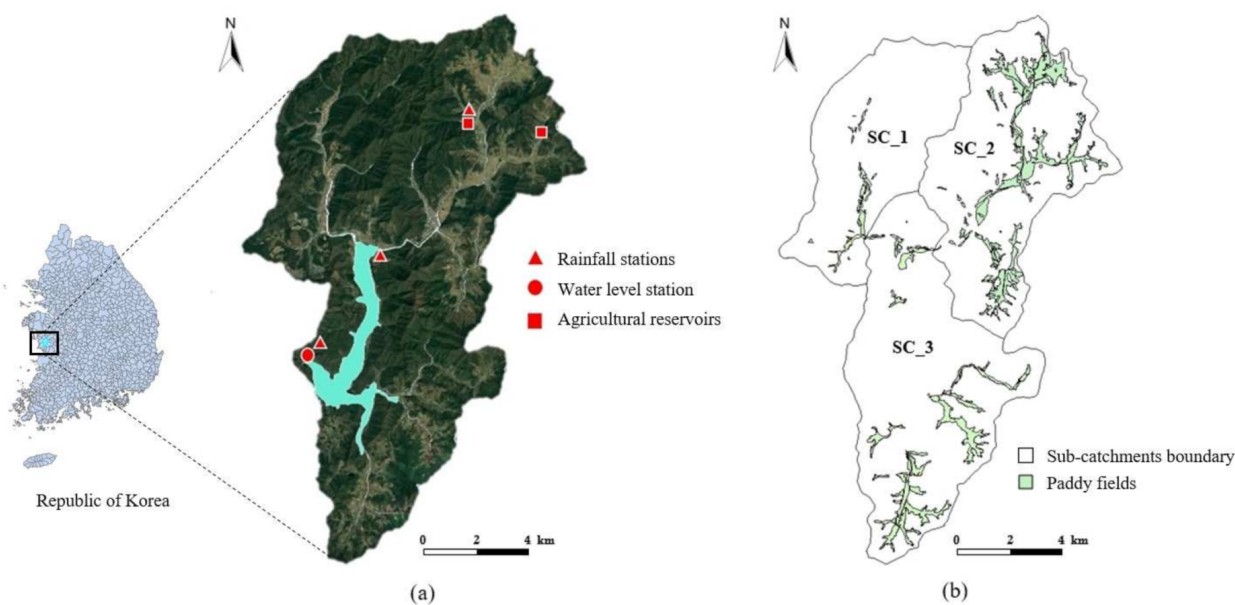

**Figure 1.** (**a**) Location of the rainfall stations, the water level station, and the agricultural reservoirs; (**b**) sub-catchments and paddy fields of Boryeong Dam catchment.

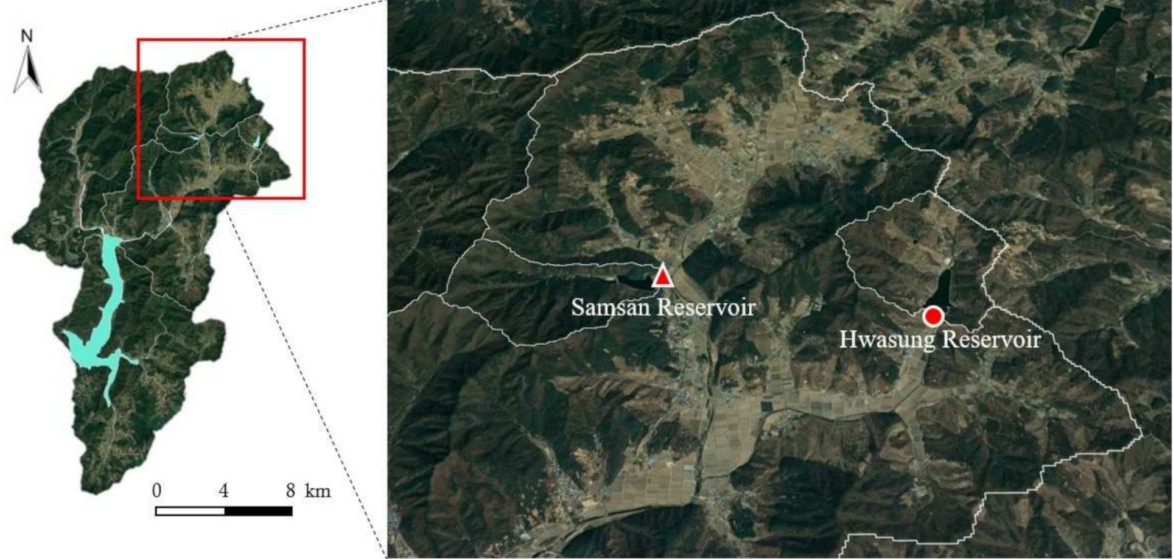

**Figure 2.** Location of the Samsan reservoir and the Hwasung reservoir in the SC_2 sub-catchment of the Boryeong Dam catchment.

**Table 1.** Geographic locations, areas, storage capacities, and embankment information of the Samsan and Hwasung reservoirs.

| Reservoir | Latitude (DMS) | Longitude (DMS) | Catchment Area (km$^2$) | Maximum Surface Area (m$^2$) | Total Storage Capacity (m$^3$) | Effective Storage Capacity (m$^3$) | Height of Embankment (m) |
|---|---|---|---|---|---|---|---|
| Samsan | 36°20′14″ | 126°43′10″ | 1.8 | 52,000 | 546,200 | 543,200 | 25.7 |
| Hwasung | 36°20′07″ | 126°45′13″ | 2.05 | 85,800 | 724,400 | 720,900 | 24.7 |

For the streamflow simulation considering the agricultural reservoirs located in the SC_2 sub-catchment, the sub-catchment was divided into 10 smaller sub-catchments according to the Samsan Reservoir catchment, Hwasung Reservoir catchment, upstream catchments, downstream catchment, and the irrigated paddy fields of each sub-catchment. The irrigated districts by two agricultural reservoirs are the target paddy fields for the analysis of soil moisture and groundwater level in this study. The irrigated paddy field by Samsan reservoir is denoted as P_Samsan node; the irrigated paddy field by Hwasung reservoir is denoted as P_Hwasung node in the CAT system. The reservoir nodes each have the upstream sub-catchments (S1 and S2), and paddy fields in S1 and S2 were also created (P1 and P2). In addition, the downstream sub-catchments of the P_Samsan and P_Hwasung were created as S3 and P3, respectively. The CAT system setup considering the division of sub-catchments and paddy fields is shown in Figure 3.

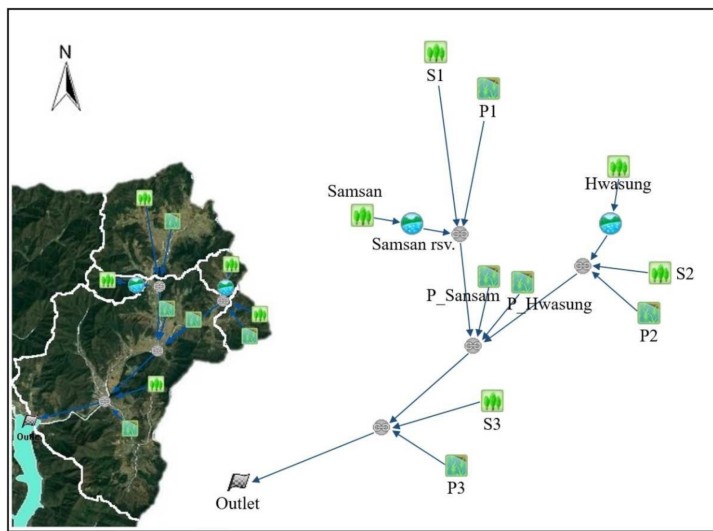

**Figure 3.** CAT system setup with reservoir sub-catchments, upstream sub-catchments of each reservoirs, downstream sub-catchment, and paddy fields considering Samsan and Hwasung Reservoirs.

The Samsan Reservoir and Hwasung Reservoir nodes were created to input the agricultural water supply data of the reservoirs. The areas, the slopes, and the ratio of impervious area for each sub-catchment were calculated using GIS processes. The paddy fields supplied the agricultural water from the Samsan Reservoir and Hwasung Reservoir are located in the downstream catchment outside each reservoir's sub-catchments. Detailed descriptions of sub-catchments and nodes are shown in Table 2.

**Table 2.** Description of each node of CAT modeling of SC_2 sub-catchment of Boryeong Dam catchment.

| Node | Description | Node | Description |
|---|---|---|---|
| Samsan | Samsan reservoir catchment | S1 | Upstream catchment of P_Samsan |
| Samsan rsv. | Samsan reservoir | P1 | Paddy field in C1 |
| P_Samsan | Irrigated paddy field by Samsan reservoir | S2 | Upstream catchment of P_Hwasung |
| Hwasung | Hwasung reservoir catchment | P2 | Paddy field in C2 |
| Hwasung_rsv. | Hwasung reservoir | S3 | Downstream catchment |
| P_Hwasung | Irrigated paddy field by Hwasung reservoir | P3 | Paddy field in C3 |

Table 3 shows the areas, slopes, impervious area ratio, and the amount of groundwater withdrawals by pumping in each sub-catchment and paddy field. About 90% of the groundwater withdrawals in the Boryeong Dam catchment is used for agricultural purposes, and the remaining 10% is used for domestic and industrial water demands. Agricultural water consumptions in the sub-catchments were input to the irrigated paddy fields of each sub-catchment, and water consumptions for domestic and industrial demands were input to the remaining forest sub-catchments, except for paddy fields.

**Table 3.** Area, slope, impervious area ratio, and annual amount of groundwater abstraction in sub-catchments and paddy fields.

| Sub-Catchments | Samsan | Hwasung | S1 | S2 | S3 |
|---|---|---|---|---|---|
| Area (km$^2$) | 1.8 | 2.05 | 12.25 | 30.70 | 2.21 |
| Slope (%) | 0.354 | 0.282 | 0.281 | 0.277 | 0.302 |
| Impv_a (%) | 1.02 | 2.46 | 5.48 | 7.84 | 3.11 |
| GW_pump (m$^3$/day) | 19.58 | 22.30 | 133.24 | 333.93 | 23.98 |
| **Paddy Fields** | **P_Samsan** | **P_Hwasung** | **P1** | **P2** | **P3** |
| Area (km$^2$) | 0.8 | 0.80 | 2.5 | 0.20 | 3.12 |
| Slope (%) | 0.122 | 0.110 | 0.095 | 0.088 | 0.079 |
| Impv_a (%) | 0.00 | 0.00 | 0.00 | 0.00 | 0.00 |
| GW_pump (m$^3$/day) | 1384.91 | 1388.37 | 4327.84 | 342.77 | 5401.14 |

Impv_a refers to the ratio of impervious area; GW_pump refers to observed daily groundwater abstraction.

## 2.2. CAT Model

CAT is a physical parameter-based and distributed hydrological model that allows the quantitative evaluation of the long- and short-term water cycles of the catchment. CAT is a node- and link-connecting model designed to estimate the hydrological components, such as runoff, infiltration, soil moisture content, evapotranspiration, and baseflow, for each spatial unit. It divides the hydrological cycle process into pervious and impervious areas. The model simulation can be conducted with minute, hour, day, month, and year time step data. The basic concept of the model is based on the unconfined aquifer and single soil layer assumptions. The model categorizes the incoming rainfall into falling on pervious, impervious, and paddy field areas, yielding the surface flow, infiltration, or evapotranspiration (Figure 4). The major physical parameters required for the initial simulation are area, slope, soil type, land use, aquifer, and river information. The basic concept of the model is based on the unconfined aquifer and single soil layer assumptions [27]. In addition, the CAT simulation can be carried out taking into account the actual agriculture situation, such as water supply from agricultural reservoirs, irrigation period, and levee height in the paddy fields of the catchment.

The CAT model has been applied in various hydrological studies for different catchments. Jang et al. [28] analyzed the long-term hydrological responses of agricultural reservoirs in the Idong catchment; Jang et al. [29] assessed the future climate change impacts on the hydrological components in the Gyeongan–Cheon River Basin. Birhanu et al. [30] analyzed and compared the results of five hydrological models, including CAT, applying them in 10 catchments of Korea. Choi et al. [31] carried out a short-term CAT runoff simulation and a sensitivity analysis of soil parameters using three infiltration methods provided in CAT. Lee and Cho [32] analyzed the hydrological cycle in four catchments in Ulsan City using CAT. Miller et al. [33] evaluated stormwater runoff characteristics according to the transformation of rural landscapes into peri-urban areas in the U.K. using CAT.

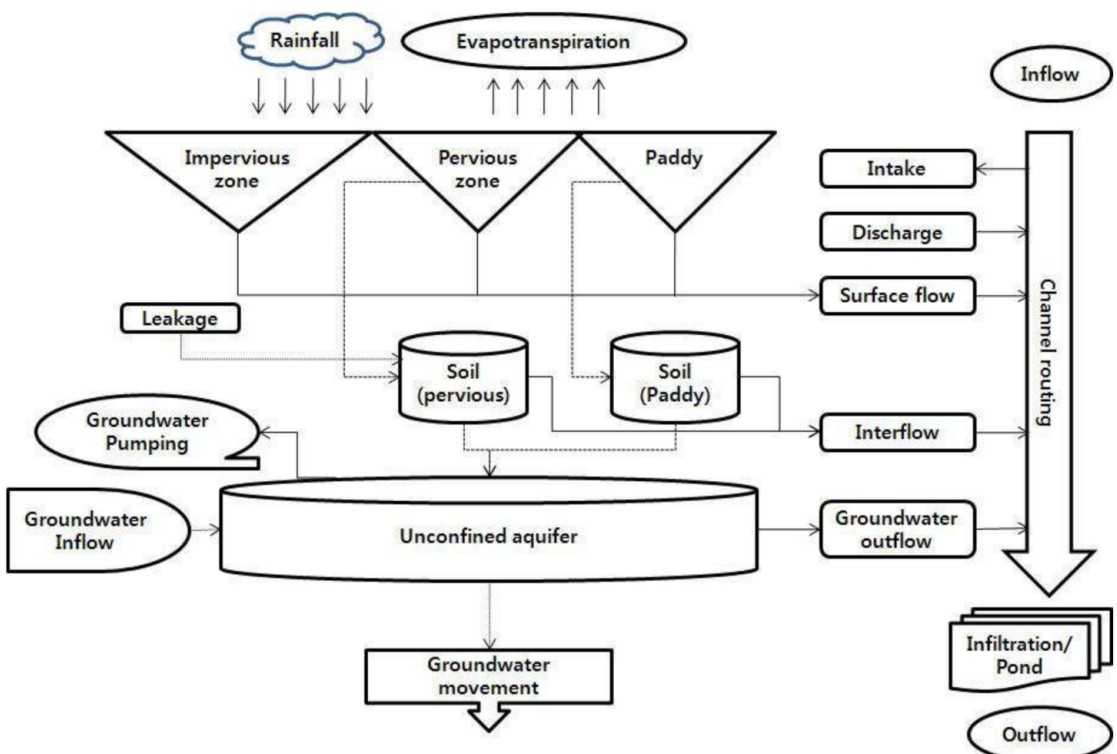

**Figure 4.** Schematic diagram of water cycle process in CAT model [34].

### 2.3. Water Cycle Processes of Reservoir and Paddy Field in CAT

In the CAT model, the storage facility node (or reservoir node) can be applied as an online reservoir when the reservoir is located within the stream channel, or as an offline reservoir when the reservoir is located outside the stream channel. In this study, the reservoir node was applied as an online reservoir by reflecting the field situations of the Boryeong Dam catchment. The reservoir node considers the amount of evaporation from the surface of the reservoir and intake amount of water to calculate the amount of outflow discharged through the outlet (Figure 5).

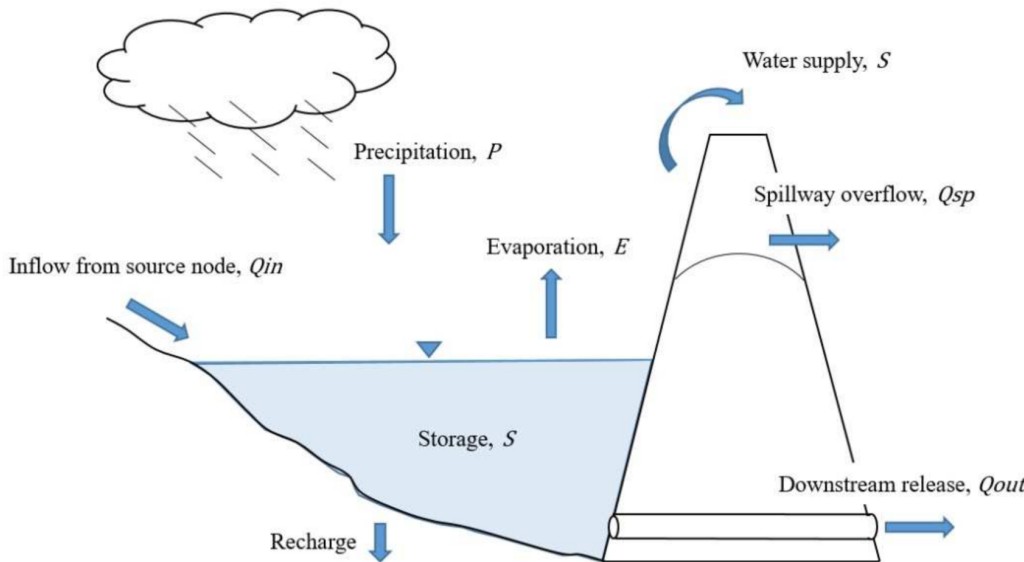

**Figure 5.** Schematic diagram of hydrological cycle in reservoir in CAT model.

The governing equation is:

$$\frac{dS}{dt} = Qin - Qout - Qsp + P - E \tag{1}$$

where $S$ is the storage volume (m$^3$), $Q_{in}$ is the inflow to the reservoir (m$^3$/s), $Q_{out}$ is the outflow of the reservoir (m$^3$/s), $Q_{sp}$ is the spillway overflow (m$^3$/s), $P$ is the precipitation to the surface of reservoir (mm), and $E$ is the evaporation from the surface of the reservoir (mm) [34].

To simulate the runoff process in the paddy field (Figure 6), the soil and groundwater layers were divided in the same way as the pervious area of the watershed. Artificial drainage facilities can be included in the soil layer to accommodate underground culvert drainage and pipe drainage; however, only surface drainage by levee height was considered in this study according to the field situation of the catchment. Surface drainage occurs when the ponding depth is greater than the height of the surface drain water threshold.

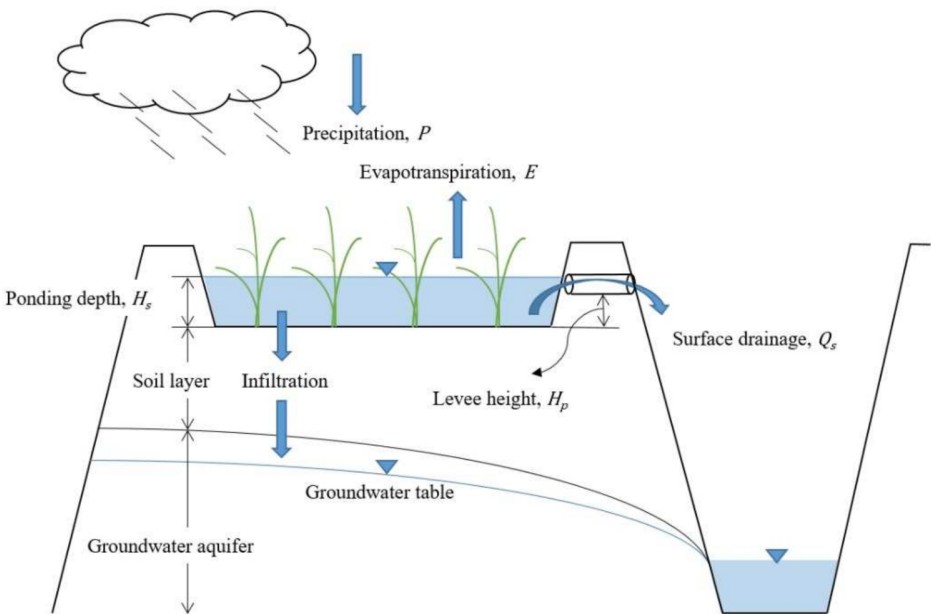

**Figure 6.** Schematic diagram of the hydrological cycle in the paddy field in CAT model.

The surface drain equations are:

$$Q_s = \alpha\sqrt{H_s - H_p}\,(H_s > H_p) \tag{2}$$

$$Q_s = 0 \quad (H_s \leq H_p) \tag{3}$$

where $Q_s$ is the discharge from the surface (m$^3$/s), $\alpha$ is the drainage coefficient of the surface drain levee in the paddy (mm$^{0.5}$/h), $H_s$ is the ponding depth of the paddy (m), and $H_p$ is the height of the surface drain levee of the paddy (m) [34].

The hydrological interrelation between surface water and groundwater was calculated according to Darcy's Law, with flows based on the hydraulic conductivities, river stage, and groundwater level. The interrelation between the river level and the groundwater level was calculated using Equations (4) and (5). Equation (5) was applied when the river level was higher than the groundwater level in the vicinity; otherwise, Equation (4) was applied.

$$Q_r = K_{sr}A_r \tag{4}$$

$$Q_r = K_{sr}\left(\frac{h - H_r}{b_r}\right)A_r \tag{5}$$

$$Q_g = K_{sr} \frac{\partial h}{\partial x} \cdot l \cdot T \tag{6}$$

$$Q_{in} - Q_{out} = A \cdot S \frac{dh}{dt} \tag{7}$$

where $Q_r$ is the inflow into the river or recharge from the river (m³/s); $Q_g$ is the groundwater flow (m³/s); $Q_{in}$ and $Q_{out}$ are the inflow and outflow of the aquifer (m³/s), respectively; $K_{sr}$ is the saturated hydraulic conductivity of the riverbed (m/s); $A_r$ is the area of the riverbed (m²); $b_r$ is the riverbed thickness (m); $h$ is the groundwater level (m); $H_r$ is the riverbed elevation (m); $\partial h/\partial x$ is the slope of the groundwater level; $l$ is the connected length between catchments (m); $T$ is the average aquifer thickness (m); $A$ is the catchment area (m²); $S$ is the storage coefficient; and $dh/dt$ is the rate of level change [34].

### 2.4. Data Collection

The daily precipitation, streamflow, and meteorological data from 2000 to 2019 were already collected for a previous study [24]. The daily precipitation data were gathered from the three rainfall gauging stations operated by K-water [23]. The daily meteorological data used for the Penman–Monteith evapotranspiration estimation, such as minimum and maximum temperature (°C), humidity (%), sunshine hours, and wind speed (m/s), were gathered from the Boryeong Meteorological Station located a relatively short distance from the Boryeong Dam catchment [21]. Evapotranspiration is a critical factor of hydrological processes, especially in arid or semiarid catchments. Generally, to choose the relevant PET (potential evapotranspiration) estimation method for hydrological modeling, researchers often recommend the Penman–Monteith method as the standard for reference the evaporation estimation, which is a physically based combination equation that is able to describe the evaporation processes [35,36]. Therefore, the Penman–Monteith method was applied to estimate PET in this study. Figure 7a shows the observed precipitation and streamflow; Figure 7b shows the simulated actual and potential evapotranspiration in 2000–2019.

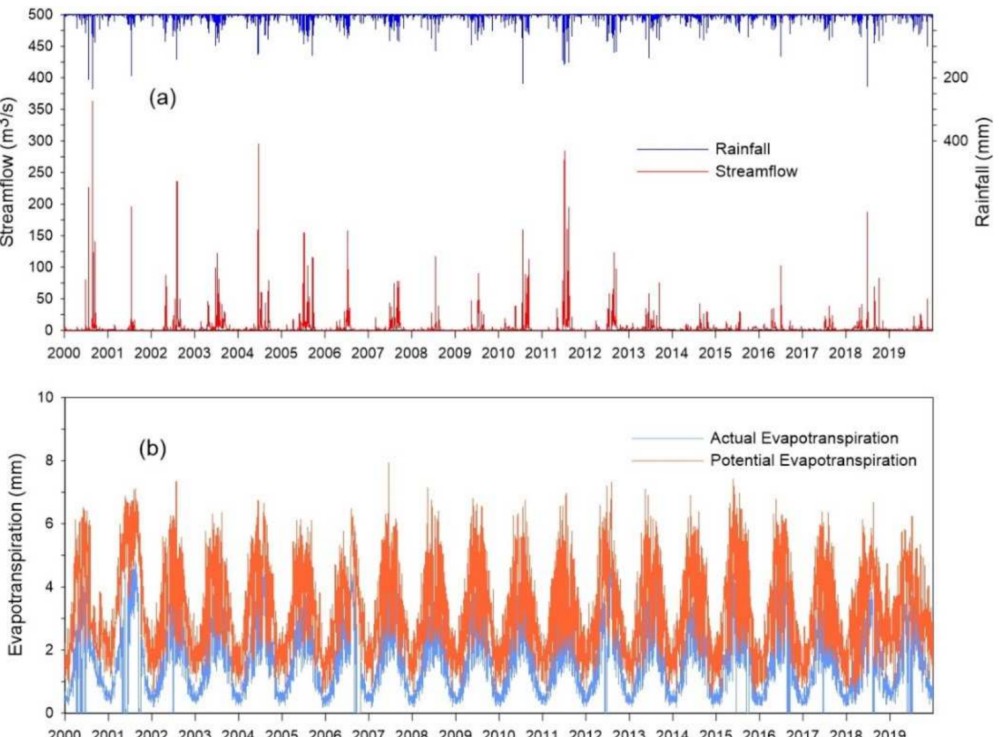

**Figure 7.** (**a**) Observed rainfall and streamflow data for 2000–2019; (**b**) estimated potential and actual evapotranspiration in the Boryeong Dam catchment.

The monthly agricultural water supply data from 2012 to 2018 were collected [5] as input data for agricultural reservoirs. The daily observed water storage data for Samsan and Hwasung reservoirs [5] were collected from May 2012 to December 2018 to compare with the simulated reservoir storage. For groundwater pumping data, the annual data for 2008–2017 were collected and applied [23]. Water balance analysis of the SC_2 sub-catchment was performed for 2012–2017, which included all data. The soil parameter data were extracted using the 1:25,000 scale soil map provided by the Rural Development Administration [37]. The land use map of the Ministry of Environment [20] was applied to classify forest area and paddy field area of the catchment. All model input data for CAT simulation were input by dividing the catchment into pervious and impervious areas. Most of the agricultural water is supplied during the irrigation period, from April to September. In 2018, both reservoirs supplied agricultural water from May to August. The monthly amount of agricultural water supply usually varies depending on the rice growth process in paddy fields. In the Samsan Reservoir and Hwasung Reservoir, agricultural water supply was the highest in May and lowest in April (Figure 8).

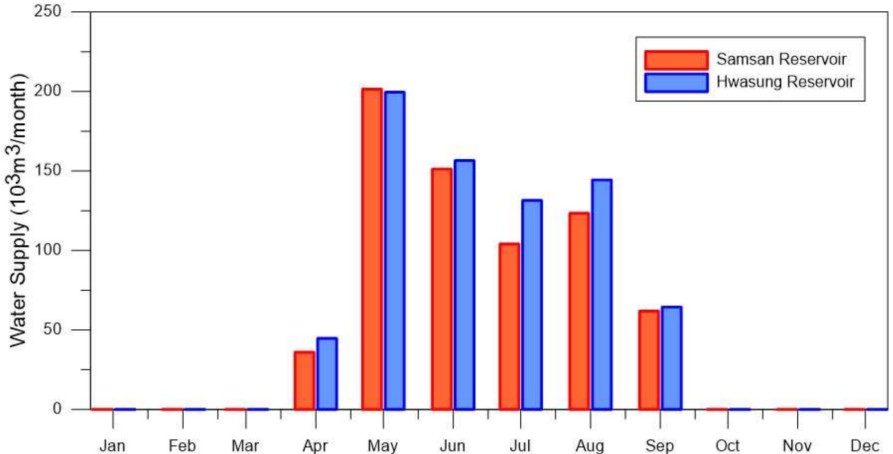

**Figure 8.** Average monthly amount of agricultural water supply of Samsan Reservoir and Hwasung Reservoir.

In April, the total amounts of agricultural water supply of the Samsan Reservoir and Hwasung Reservoir were 36,007 $m^3$ and 44,670 $m^3$, respectively, and the total agricultural water supply amounts in May were 201,431 $m^3$ and 199,541 $m^3$, respectively, indicating that the largest consumption of agricultural water is in May when rice planting begins. The total agricultural water supply by year of each reservoir is shown in Table 4.

**Table 4.** Annual agricultural water supply of the Samsan and Hwasung reservoirs.

| | | | | | (Unit: $10^3$ $m^3$) |
|---|---|---|---|---|---|
| Year | Samsan Rsv. | Hwasung Rsv. | Year | Samsan Rsv. | Hwasung Rsv. |
| 2012 | 944.8 | 953.33 | 2016 | 338.63 | 642.17 |
| 2013 | 814.6 | 822.8 | 2017 | 685.64 | 415.72 |
| 2014 | 782.6 | 899.33 | 2018 | 625.26 | 579.79 |
| 2015 | 553.93 | 873.49 | Avg. | 677.92 | 740.95 |

In the case of Samsan Reservoir, the annual average of 677,920 $m^3$ of the agricultural water was supplied to the irrigation district from 2012 to 2018; for the Hwasung Reservoir, an annual average of 740,950 $m^3$ of agricultural water was supplied.

The daily observed storage data, the monthly agricultural water supply data, and the specification of the Samsan and Hwasung reservoirs were collected from the Rural Agricultural Water Resource Information System (RAWRIS) [38] managed by the Korea Rural Community Corporation [5].

### 2.5. Model Performance Indicators

The multi-objective functions used for evaluating model performances in this study were the Kling–Gupta Efficiency (KGE), Nash–Sutcliffe Efficiency (NSE), LogNSE, the determination coefficient ($R^2$), and the root mean square error–observations standard deviation ratio (RSR).

NSE is a normalized indicator that determines the relative magnitude of the residual variance compared to the observed data variance; however, it has the disadvantage that the peak value is often overestimated or the valley value is often underestimated by comparing the variance of the error and the variance of the observed values in a relative volume. LogNSE is the logarithm-transformed NSE often applied to put more importance on low flow simulations in hydrological modeling [39]. KGE is an index that aims to overcome the disadvantages of NSE by considering the correlation coefficient and the mean as well as variance. RSR was developed to standardize RMSE using observations in standard deviation, combining the error index and the additional information [40].

The values of the model performance evaluation index closer to 1, excluding RSR, indicate better model performance; in the case of RSR, the closer to 0, the better the model performance. When the NSE > 0.5 and RSR < 0.6, it can be judged as a satisfactory model performance [41].

$$KGE = 1 - \sqrt{\left(R^2[Q_{obs}, Q_{sim}] - 1\right)^2 + \left(\frac{SD[Q_{sim}]}{SD[Q_{obs}]} - 1\right)^2 + \left(\frac{M[Q_{sim}]}{M[Q_{obs}]} - 1\right)^2} \tag{8}$$

$$NSE = 1 - \frac{\sum(Q_{obs} - Q_{sim})^2}{\sum(Q_{obs} - Q_{mean})^2} \tag{9}$$

$$R^2 = 1 - \frac{\sum(Q_{obs} - Q_{sim})^2}{\sum(Q_{obs} - \overline{Q_{obs}})^2} \tag{10}$$

$$RSR = \frac{RMSE}{SD[Q_{obs}]} = \frac{\sqrt{\sum(Q_{obs} - Q_{sim})^2}}{\sqrt{\sum(Q_{obs} - Q_{mean})^2}} \tag{11}$$

where $Q_{obs}$, $Q_{sim}$, and $Q_{mean}$ are the observed, simulated, and mean observed streamflow, respectively. SD and M are the standard deviation and mean, respectively.

In this study, the daily simulated streamflow was compared to the daily observed streamflow for the CAT model performance evaluation.

## 3. Results and Discussions

### 3.1. Model Performance of CAT

The model performance of the CAT simulation for the SC_2 sub-catchment of the Boryeong Dam catchment was evaluated by model performance indicators. As there are no observed streamflow data in the SC_2 sub-catchment, the streamflow data were estimated by dividing the total streamflow from the outlet of the entire Boryeong Dam catchment by the area ratio of the SC_2 sub-catchment according to the drainage area ratio method [42].

The CAT runoff simulation was carried out during the period of 2012–2017, including both the agricultural water supply data and the groundwater abstraction data of Samsan and Hwasung reservoirs. As a result of the daily runoff simulation, the simulated streamflow was well-matched to the observed one (Figure 9a). The KGE, NSE, $R^2$, and RSR values were 0.88, 0.79, 0.80, and 0.42, respectively, indicating the satisfactory accuracy of the CAT model (Figure 9b). The simulated streamflow tended to be slightly underestimated against the observed one when the streamflow was small, and it tended to be slightly overestimated when the streamflow was large.

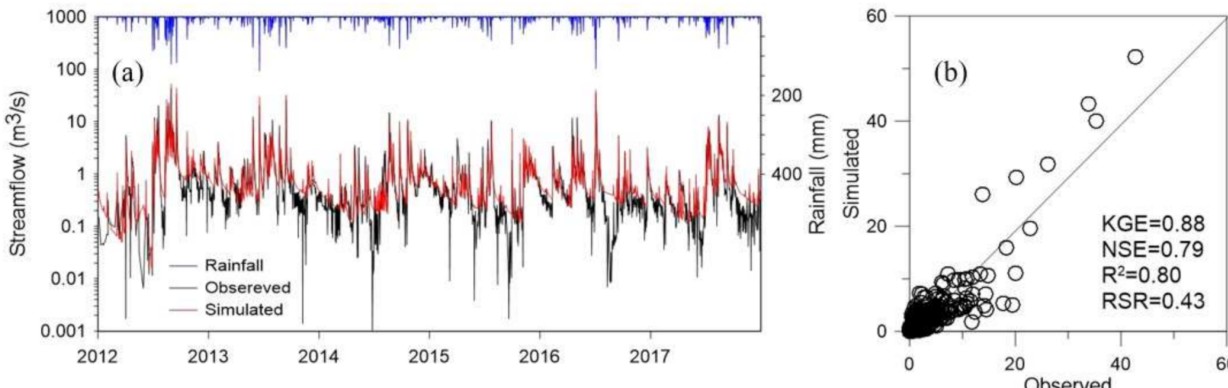

**Figure 9.** (**a**) Comparison of the estimated observed streamflow and the CAT simulated streamflow. (**b**) Scatter plot of observe and simulated streamflow and the model performances by statistical indicators.

The observed storage data of the Samsan Reservoir and Hwasung Reservoir for May 2012 to December 2018 [5] were compared with the CAT simulated storage (Figure 10). Comparing the sum of observed and simulated daily storages during the study period, the simulated storage of the Samsan Reservoir was 99.84% of the observed storage, and the simulated storage of the Hwasung Reservoir was 99.1% of the observed storage. The CAT simulated storage was underestimated or overestimated by year; however, the annual inflow calibration was not performed in the Samsan and Hwasung nodes, as this study aims to analyze the water balance through long-term runoff simulation and the soil moisture and groundwater level in the paddy fields, according to the agricultural water supply of the reservoir.

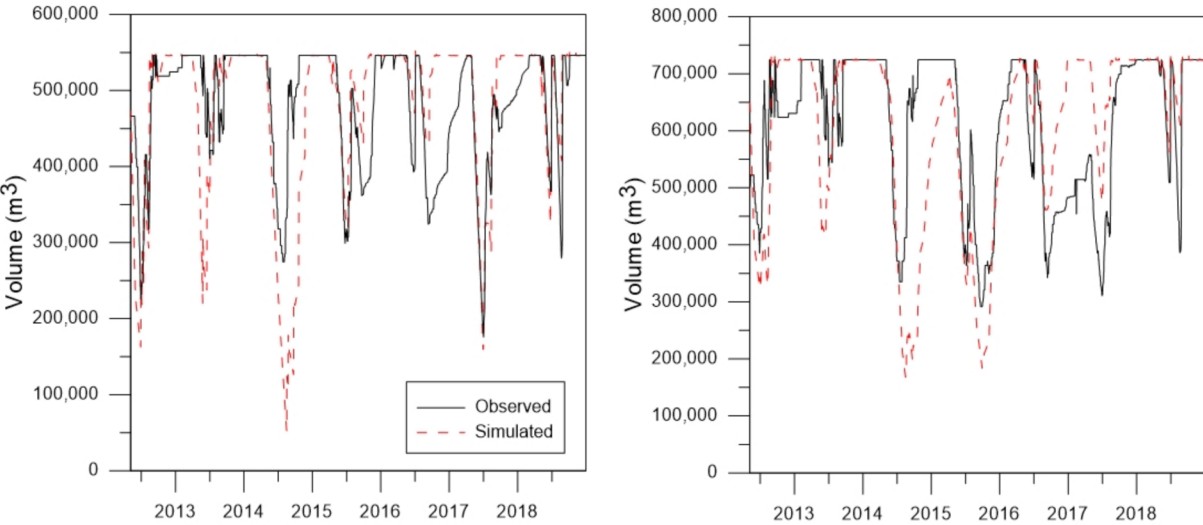

**Figure 10.** Comparison of the observed and the simulated water storage of Samsan Reservoir and Hwasung Reservoir from May 2012 to December 2018.

### 3.2. Water Balance Analysis

To analyze the change in water cycle in the catchment according to the presence or absence of the agricultural reservoirs, two models with and without the reservoirs were constructed, and the simulated water balances were compared. The model without reservoirs and the model with reservoirs were constructed with the same conditions, except for the amount of water intake from the agricultural reservoirs.

Figure 11a shows the simulated water balance of the SC_2 sub-catchment to which agricultural reservoirs were applied. Figure 11b shows the simulated water balance to

which agricultural reservoirs were not applied. When the agricultural reservoirs were applied in the CAT simulation, the amount of inflow due to the supply of agricultural water from the reservoirs was considered. The results showed that the total amount of streamflow and the groundwater recharge increase because the reserved water from Samsan and Hwasung reservoirs was supplied to the soil surface of the catchment.

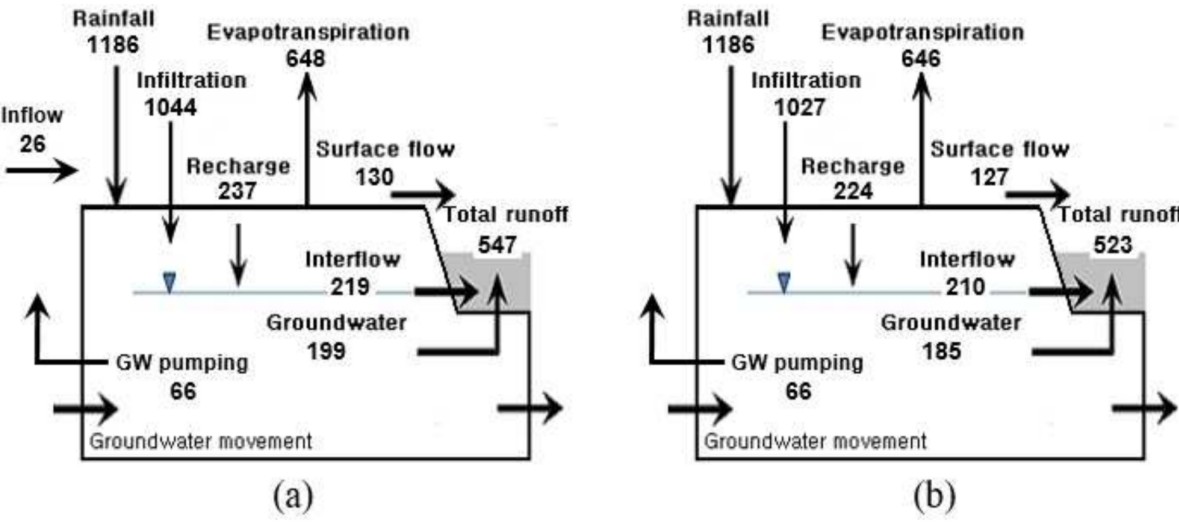

**Figure 11.** (**a**) Simulated water cycle of the CAT model with application of agricultural reservoirs. (**b**) Simulated water cycle of the CAT model without application of agricultural reservoirs.

By applying the agricultural reservoirs, the total amount of runoff in the Boryeong Dam catchment increased by about 4.72% over the study period due to the supply of agricultural water. The surface runoff, interflow, and baseflow increased by 1.27%, 4.44%, and 6.81%, respectively, and the amount of groundwater recharge increased by about 5.67% (Table 5). Therefore, as the amount of total runoff and groundwater recharge of the catchment increase according to the application of the agricultural reservoirs in the CAT simulation, it can be effective in improving the water cycle soundness of the catchment by proper operation of reservoirs.

**Table 5.** Increase rate of amount of runoff and groundwater recharge after application of agricultural reservoirs compared to the amount of runoff and groundwater recharge before the application of agricultural reservoirs.

| | | | | | (Unit: %) |
|---|---|---|---|---|---|
| Year | Total Runoff | Surface Runoff | Interflow | Baseflow | Recharge |
| 2012 | 3.71 | 3.01 | 2.98 | 5.57 | 3.95 |
| 2013 | 4.35 | 1.84 | 3.81 | 6.39 | 4.95 |
| 2014 | 7.35 | 0.36 | 6.91 | 11.40 | 8.82 |
| 2015 | 5.48 | 0.88 | 5.38 | 7.48 | 6.94 |
| 2016 | 2.96 | 0.61 | 3.20 | 4.27 | 4.13 |
| 2017 | 4.45 | 0.93 | 4.37 | 5.78 | 5.23 |
| Avg. | 4.72 | 1.27 | 4.44 | 6.81 | 5.67 |

*3.3. Soil Moisture Content of Paddy Fields*

As the soil moisture content is a hydrological component that directly affects crop growth, the accurate estimation of soil moisture content in paddy fields should be preceded in the hydrological analysis of agricultural watersheds. In this study, to analyze the effect of drought on the soil moisture content in paddy fields, the representative drought and normal years were divided and analyzed within the study period. Comparing the

drought index in Buyeo-gun [14], where the SC_2 sub-catchment is located, the SGI and SPI drought indices in 2013 were confirmed to be the normal year, and the SMI drought indices indicated frequent dryness. In 2015, SGI indicated the caution drought level, SPI indicated the extreme drought level, and SMI indicated the frequent dry or normal drought level. Therefore, 2013 was selected as the WET period and 2015 as the DRY period, representing the year of severe drought. For soil moisture content analysis, the models with and without the reservoirs were compared according to the WET and DRY periods.

Figure 12a,b show the change in soil moisture content at the P_Samsan node according to the WET and DRY periods when the Samsan Reservoir was applied. Figure 12c,d show the change in soil moisture content at the P_Hwasung node according to the WET and DRY periods when the Hwasung Reservoir was applied. The CAT simulation indicated that the soil moisture content increases during the irrigation period from April to September when the agricultural reservoirs were applied to the CAT model. In the WET period, the difference in soil moisture according to the application of the reservoirs is significant from April, when the irrigation season begins, until the end of June, when the rainy season starts. In the DRY period, the difference in the soil moisture content according to the application of the reservoirs is significantly large until November due to the lack of precipitation. There was no significant difference in the trends of soil moisture content changes in the Samsan Reservoir and Hwasung Reservoir. The analyses of the change in soil moisture content by irrigation and non-irrigation periods are shown in Figure 13.

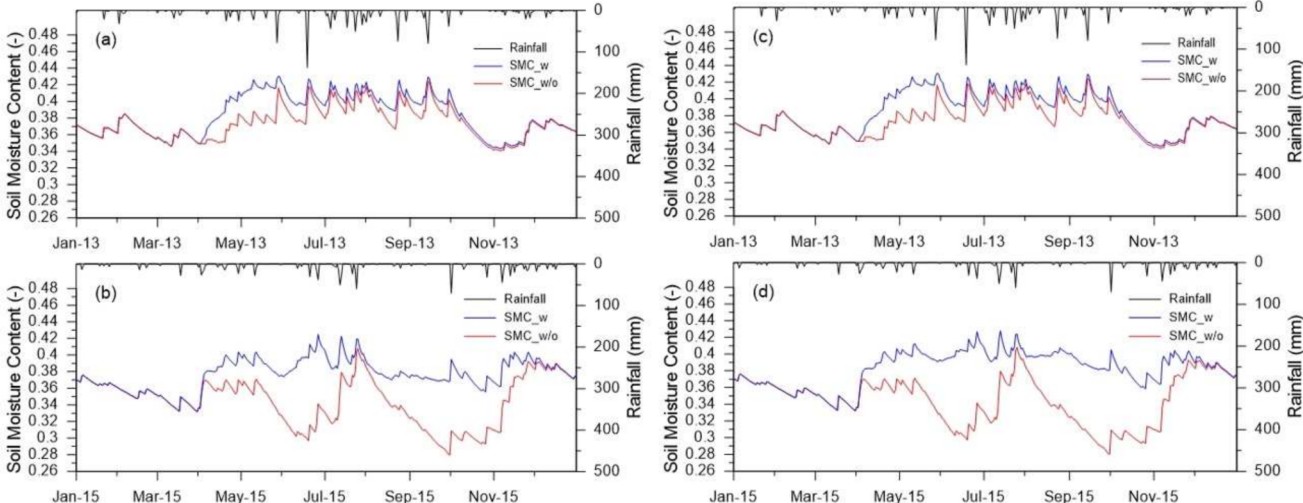

**Figure 12.** (**a**) Soil moisture content changes of P_Samsan in the WET year by applying reservoir nodes in CAT. (**b**) Soil moisture content changes of P_Samsan in the DRY year by applying reservoir nodes in CAT. (**c**) Soil moisture content changes of P_Hwasung in the WET year by applying reservoir nodes in CAT. (**d**) Soil moisture content changes of P_Hwasung in the DRY year by applying reservoir nodes in CAT.

The increase rate of the soil moisture content according to the application of the agricultural reservoirs was higher in the irrigation period than in the non-irrigation period, and was higher in the DRY period than in the WET period. In the WET period, the average soil moisture content during the irrigation period increased by 5.25% for P_Samsan and by 5.14% for P_Hwasung, according to the operation of the agricultural reservoirs. The average soil moisture content for P_Samsan and P_Hwasung during the non-irrigation period increased by 0.44% and 0.43%, respectively. In the DRY period, the average soil moisture content during the irrigation period increased by 8.91% for P_Samsan and 10.92% for P_Hwasung, according to the operation of the agricultural reservoirs. The average soil moisture content for P_Samsan and P_Hwasung during the non-irrigation period increased by 2.49% and 2.59%, respectively (Table 6).

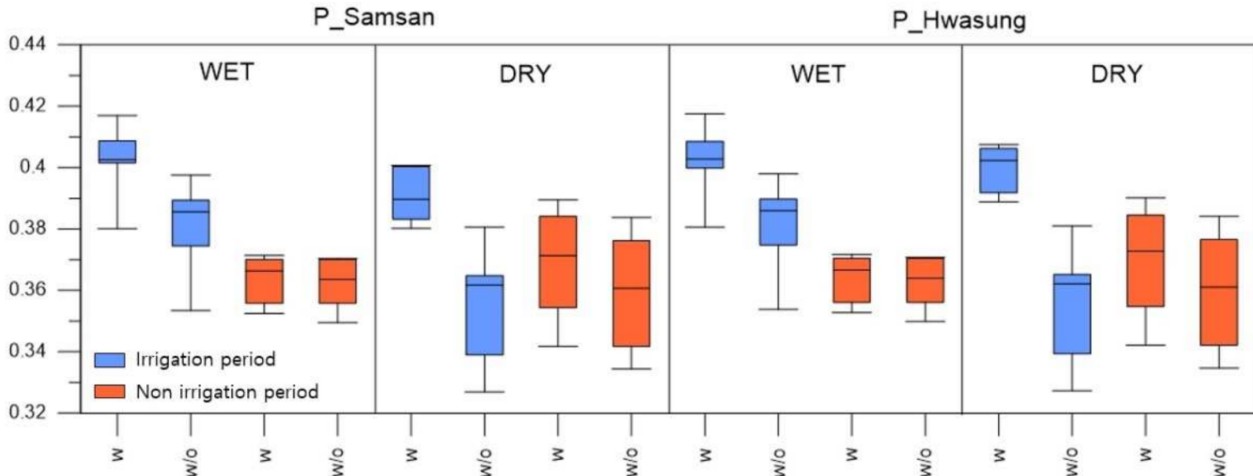

**Figure 13.** The changes in soil moisture content of P_Samsan and P_Hwasung in the irrigation period and the non-irrigation period by WET and DRY years.

**Table 6.** Average soil moisture in irrigation and non-irrigation periods and the increase rate by applying agricultural reservoirs.

| Period | | P_Samsan | | | P_Hwasung | | |
|---|---|---|---|---|---|---|---|
| | | w/o | w | Increase (%) | w/o | w | Increase (%) |
| WET | Irrig. | 0.381 | 0.402 | 5.25 | 0.381 | 0.402 | 5.14 |
| | Non irrig. | 0.362 | 0.364 | 0.44 | 0.362 | 0.364 | 0.43 |
| DRY | Irrig. | 0.356 | 0.391 | 8.91 | 0.356 | 0.400 | 10.92 |
| | Non irrig. | 0.360 | 0.369 | 2.49 | 0.360 | 0.370 | 2.59 |

Irrig. refers to the irrigation period of April to September. Non_irrig. refers to the non-irrigation period. w/o refers to the modeling without the reservoirs, and w refers to the modeling with the reservoirs.

The increase rate of soil moisture content during the DRY period was relatively higher than that during the WET period because the soil layer dried due to drought and lack of precipitation; therefore, it responded sensitively to the reservoir agricultural water supply and absorbed more water. Soil moisture is the retained water in the topsoil layer and has direct effects on the growth of rice during the irrigation period. Analyzing the changes in soil moisture content during the irrigation period in Korea from April to September, we can see that the agricultural water supplies of reservoirs significantly affected the soil moisture increase in the paddy fields, especially in the drought period. Therefore, in order to secure the soil moisture demands for agricultural activities in the paddy catchments, it is necessary to supply sufficient amounts of agricultural water through the installation of agricultural reservoirs.

### 3.4. The Groundwater Elevation in Paddy Fields

The groundwater elevation changes in P_Samsan and P_Hwasung according to the application of the Samsan Reservoir and Hwasung Reservoir were compared for the WET and DRY periods. Figure 14 indicated that the increase rate of the groundwater elevation due to the operation of the agricultural reservoirs was higher in the WET period than in the DRY period because the rate of groundwater abstraction was lower in the drought period than that in normal period in the catchment [24].

The increase rate of the soil moisture content according to the application of the reservoirs was higher during the DRY period; however, in the case of the groundwater elevation, the increase rate in the WET period was higher than in the DRY period, according to the application of the agricultural reservoirs.

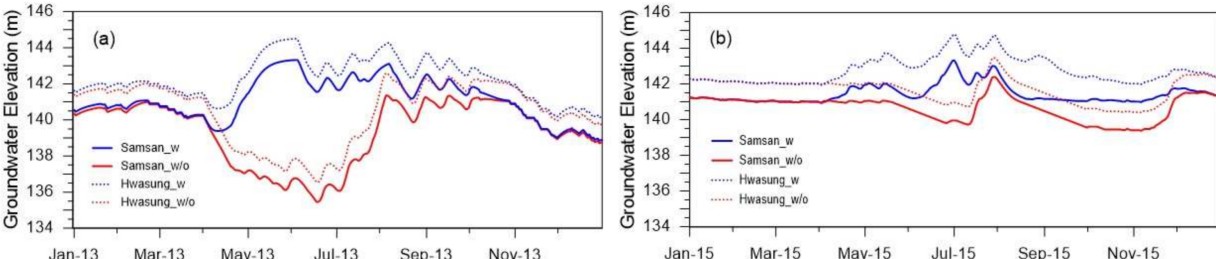

**Figure 14.** (**a**) Changes in groundwater elevation of P_Samsan and P_Hwasung, according to the application of agricultural reservoirs in the WET period. (**b**) Changes in groundwater elevation of P_Samsan and P_Hwasung, according to the application of agricultural reservoirs in the DRY period.

In the WET period, the difference in groundwater elevation reached 6.11 m in the P_Samsan node and 6.24 m in the P_Hwasung node according to the application of the agricultural reservoirs; the difference in May was the largest of the year. In the DRY period, the difference in groundwater level was 1.77 m in the P_Samsan node and 2.39 m in the P_Hwasung node according to the application of the agricultural reservoirs; the difference in June was analyzed to be the largest of the year (Table 7). During the DRY period, the total amount of groundwater consumption was smaller than in the WET period; therefore, the decrease in the groundwater elevation due to groundwater pumping was not significant in the DRY period. The amount of groundwater abstraction in 2013 was 2255 $m^3$/day and 2261 $m^3$/day for P_Samsan and P_Hwasung, respectively; in 2015, it was 457 $m^3$/day and 458 $m^3$/day, respectively. Therefore, the difference in groundwater elevation according to the application of the agricultural reservoirs was larger in the WET period than in the DRY period. If the reservoir is not operated, the groundwater level in the paddy field will decrease rapidly as irrigation begins in April. The supply of agricultural water from the reservoirs to the paddy fields seems to prevent the decrease in the groundwater elevation in irrigation period.

**Table 7.** Increased monthly groundwater elevation of P_Samsan and P_Hwasung after application of agricultural reservoirs compared to monthly groundwater elevation before application of agricultural reservoirs.

| | | | | (Unit: m) |
| --- | --- | --- | --- | --- |
| **Month** | **WET** | | **DRY** | |
| | **P_Samsan** | **P_Hwasung** | **P_Samsan** | **P_Hwasung** |
| Jan | 0.20 | 0.28 | 0.00 | 0.03 |
| Feb | 0.15 | 0.23 | 0.00 | 0.02 |
| Mar | 0.07 | 0.19 | 0.00 | 0.03 |
| Apr | 1.82 | 2.04 | 0.43 | 0.56 |
| May | 6.11 | 6.24 | 0.83 | 1.41 |
| Jun | 6.09 | 5.92 | 1.77 | 2.39 |
| Jul | 4.44 | 4.19 | 1.74 | 2.31 |
| Aug | 1.42 | 1.48 | 0.40 | 1.43 |
| Sep | 0.94 | 1.04 | 1.08 | 1.74 |
| Oct | 0.25 | 0.29 | 1.59 | 1.71 |
| Nov | 0.04 | 0.14 | 1.41 | 1.44 |
| Dec | 0.12 | 0.34 | 0.13 | 0.17 |
| Avg. | 1.81 | 1.87 | 0.78 | 1.11 |

## 4. Conclusions

The effect of agricultural reservoirs on improving the water cycle of the SC_2 sub-catchment of the Boryeong Dam catchment was evaluated by applying agricultural reservoir nodes in the CAT model, and the soil moisture content and groundwater elevation

changes in paddy fields were compared and analyzed according to the division of normal and drought periods.

As a result of the CAT runoff simulation for the SC_2 sub-catchment, the KGE, NSE, $R^2$, and RSR indicators were 0.88, 0.79, 0.80, and 0.42, respectively, indicating the satisfactory performance of the CAT model. The observed streamflow was estimated by the area ratio and total discharge at the outlet of the entire Boryeong Dam catchment. Comparing the simulated water storage of the Samsan Reservoir and the Hwasung Reservoir with the observed water storage, there is a match of more than 99% in terms of the total water storage volume during the study period; however, it was either under or overestimated by year. In this paper, as the long-term streamflow simulation for 2012–2017 is performed, the yearly calibration for the reservoirs' storage was not conducted. To analyze the water balance of the SC_2 sub-catchment, the results were compared by simulating the model that applied the agricultural reservoirs in the catchment and the model that did not apply the agricultural reservoirs. As a result of the streamflow simulation, the total amount of streamflow and the groundwater recharge of the catchment increased when considering the agricultural water supply, according to the application of the agricultural reservoirs. The total streamflow increased by 4.92% per year on average, and the groundwater recharge increased by 5.96% per year on average. In addition, to analyze the soil moisture content and groundwater elevation in the reservoirs irrigated districts, WET and DRY periods were selected during the study period according to the SGI, SPI, and SMI drought indices. Analyzing the soil moisture content in WET and DRY periods according to the agricultural reservoirs' application, it is evident that the soil moisture content in the irrigation period from April to September increased when the reservoir was applied. The increase in soil moisture content was significant during the DRY period and was analyzed to be higher during the irrigation period than the non-irrigation period. In particular, the high rate of increase in the soil moisture content during the DRY period is likely due to the sensitive response of the dried soil layer to the water supply of the agricultural reservoirs. On the other hand, it was analyzed that the increase rate of the groundwater elevation with the application of the agricultural reservoirs was higher in the WET period than the DRY period. During the drought period, the groundwater elevation decreased significantly as irrigation began in April, and the groundwater elevation decreased significantly during the drought period due to groundwater pumping. It was analyzed that the reservoir intake prevents the decrease in the groundwater elevation in paddy fields. There were differences in the groundwater elevation of about 6 m depending on the reservoirs during the WET period.

Therefore, the installation of agricultural reservoirs has the effect of improving the water cycle of the catchment by increasing the streamflow and groundwater recharge, the soil moisture content, and the groundwater elevation. In particular, the soil moisture content in paddy fields provides the water necessary for crop growth during the irrigation period; therefore, the increase in soil moisture content according to the supply of agricultural reservoirs is meaningful. Due to the application of the reservoirs, the soil moisture content during the irrigation period in the DRY period increases by about 10%, indicating that the impact of the agricultural reservoir on the irrigated districts is significant. As the agricultural water demands mainly depend on the supply of groundwater pumping and reservoir intake, accurate analyses of the water cycle, soil moisture content, and groundwater level in the watershed through the hydrological model should precede the designing of relevant agricultural reservoirs for the adequate conservation of groundwater.

**Author Contributions:** Conceptualization, S.P. and H.K.; Methodology, S.P., H.K., C.J., and D.K.; Formal analysis, S.P.; Investigation, S.P. and C.J.; Resources, S.P. and C.J.; Data curation, S.P. and H.K.; Writing—original draft preparation, S.P.; Writing—review and editing, H.K.; Supervision, H.K. and C.J.; Project administration, H.K.; Funding acquisition, H.K. All authors have read and agreed to the published version of the manuscript.

**Funding:** This research was funded by the Korea Institute of Civil Engineering and Building Technology, grant number 20210194-001.

**Institutional Review Board Statement:** Not applicable.

**Informed Consent Statement:** Not applicable.

**Data Availability Statement:** This study did not include any publicly available datasets.

**Conflicts of Interest:** The authors declare no conflict of interest.

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
