# Peer review of "Hydrological Effects of Agricultural Water Supplies on Paddy Fields using Surface–Groundwater Integrated Model"

_water, doi:10.3390/w14030460_

Round 1

Reviewer 1 Report

This article uses the CAT model to simulate the runoff in the Boryeong Dam watershed. On the whole, the article is well written, but there are still some issues that need to be revised. It is recommended to make major revisions before accepting it.

  1. Line 101-108, some literature should be cited in the introduction to highlight the superiority of the CAT model.
  2. Table 1. It is recommended to supplement the geographic location information such as the latitude and longitude of the studied area.
  3. Line 167-172, please elaborate on the basis for dividing sub-catchments and paddy fields.
  4. The author needs to adjust the structure of the article. I think the data collection of the article is based on the CAT model. It is recommended that this part be placed before the data collection.
  5. There are too many figures and tables in this article. Please re-integrate the figures and tables to reduce unnecessary tables or pictures. And some figures are blurry, please increase the resolution of the picture and re-produce the picture.
  6. The author's analysis of the article is very thorough, but most of the data is a pile of results, lacking a certain amount of discussion. For example, in the process of reviewing the article, I think the innovative point of the article should be to obtain some useful data through the application of the CAT model, but these data are not available from other models, or the fitting accuracy is not up to the requirement. This can be discussed in this direction.

Reviewer 2 Report

The manuscript by Park et al. presents an interesting comparison of soil moisture and groundwater level using CAT simulation both with and without agricultural reservoirs in one sub-catchment of Boryeong Dam. It would fall within the scope of this ‘Water’ journal. However, I regret to inform you that there are many concerns remaining with the description in all parts of your manuscript.

Personally, I was confused when reading the paper. Most important thing is that there are lack of written description of ‘Introduction’, and ‘Discussion’ parts. In the introduction part of your manuscript, I could not find motivation and necessity throughout deeply scope of your study. The introduction should briefly place the study in a broad context and highlight why it is important. It should define the purpose of the work and its significance, including specific hypotheses being tested. The current state of the research field should be reviewed carefully, and key publications cited. Please highlight controversial and diverging hypotheses when necessary. Briefly mention the main aim of the work and highlight the main conclusions. Keep the introduction comprehensible to scientists working outside the topic of the paper.

There is no discussion part in this manuscript. Why didn't you compare with previously reported results in the manuscript deeply? Information including results from previous studies is not well discussed. In addition, the various pieces of description are not well connected. The authors need to thoroughly go through these points (including both the content and readability of the paper) to make it more in-depth discussion.

You should have your manuscript edited by a professional English editing company. There are many unnecessary prepositions and conjunctions. Ambiguous expressions are often used (like ' analyze’, ’evaluate’... ). In addition, there are sometimes sentences that are not in the usual English paper style.

Here are some more comments (not including everything but examples) to illustrate my point:

Lines 2-3: The title should be improved. Do you need 'CAT' in the title? I think this result is not applicable everywhere, but in a limited area. You should put the target area in the title.

Line 11 ‘Therefore’: I think it's unnecessary.

Line 19 ‘evaluate the effect’: it is ambiguous expressions. You should use more clear expression.

Line 49 ’17,516’: It would be easier to understand if there was a breakdown of the size (capacity, area, etc.) of the dam.

Lines 52-53 ‘4th Long-term Comprehensive Water Resource Plan’: What's this? If you put this here, you need to at least explain what this shows.

Lines 54-55 ‘the operation of agricultural reservoirs’: What is the size of the target agricultural reservoir here?

Lines 56-68 and 69-87: The connection and relevance between the two paragraphs is unclear. Please describe it clearly.

Line 61 ‘analyze’: it is ambiguous expressions. You should use more clear expression.

Line 69 ‘crops’: does it mean paddy? You should change to ‘paddy’.

Line 76 ‘(RDI)’, Line 79 ‘(IADI)’, and Line 80’(ADAM)’: These abbreviations are not necessary, as they are not used in subsequent text.

Line 84 ‘SPI and SGI’: You should describe the full name, not the abbreviation. You should also clearly describe what makes these indices different from others and what each index represents.

Lines 83-87 ‘In this study … each periods’: This is where the work of this study suddenly comes into play. It is unclear why authors decided to do this work. Please describe how you came to this study from a review of past studies and the need for this work clearly to the reader.

Lines 86-87 ‘soil moisture … each periods’: So far, there is no review of the relationship between soil moisture and groundwater. A detailed explanation of these relationships is needed.

Line 88 ‘Researches’: Where does it refer to the target? Around only Boryeong Dam Catchment? Or in Kore?? If it is no target area, there are too few reviews.

Lines 101-108 ‘In order to … analyzed.’: If in this study it is necessary hydrologic modeling taking into account streamflow, changes in evapotranspiration, direct runoff, and baseflow in catchment unit, a thorough review of hydrological models on a catchment basis should be done. A detailed review, including studies from other regions, should be done to clarify what is needed in this study. It should clearly state the reasons for using CAT in the study.

Lines 110-129: How about describing cropping period or crop calendar of the paddy field? In this region, one crop? Or second crop? What are main rice cultivar? Are there any feature of rice paddies that differ from those in other regions? How do people use groundwater and water from reservoirs in your paddy fields? Please describe the feature including order of supply water. Please describe the feature of the groundwater extraction method. Is groundwater extraction by pumping? Manual or Electric? By Private or Group?

Line 111 ‘Boryeong Dam’: What are the reasons for focusing on this dams? What role and characteristics does this dam have for Korea?

Lines 115-116 ‘export and import water supply system’: What does this mean?

Lines 122-124 ‘About 8.11% … forest areas [20]’: What does the remaining 8% indicate?

Line 131 ‘Figure 1’: It is unclear where in Korea this map is. Please show us the exact location. The text in the legend is blurry. What does the white line in Figure 1(a) represent?

Lines 132-133 ‘the meteorological station’: Where on the map are the meteorological station?

Line 147 ‘Figure 2’: Why don't you add (a), (b), and (c) to each figure as shown in Figure 1(a) and 1(b)? References should be provided for land use map and slope.

Line 147 ‘land use map in Figure 2’: I feel the agricultural area in land use map of Figure 2 is larger than that of paddy field in Figure 1(b). Are there any crops than paddy fields?

Line 154 ‘SC_2 sub-catchment’: If only SC_2 is the target area, why do you need to show SC_1 and SC_3? Why not just show a map of the location of SC_2 for both Fig1 and Fig2?

Line 161 ‘Figure 3’: What is the significance of the Figure 3? How about you make Figure1 and Figure 3 into one diagram? What do the white lines on the map indicate? The boundary line of SC_2 sub-catchment should be shown in the figure.

Line 173 ‘Figure 4’: The figure is very blurry. What is the 'Climate' shown in Figure 4?

Line 183 ‘Table 2’: Each region seems to be able to draw the correct boundaries in the map of Figure 4 as well as the node.

Line 185 ‘Slope’: Is 'Slop (%)' average slope for each region?

Line 185 ’GW_pump’: Are there observed values for groundwater extraction? Please describe correctly what kind of data.

Line 195 ‘Boryeong Dam … Samsan station’: Are they the same as the three points shown in Figure 1? If so, the station name in Figure 1 should be shown. However, if SC_2 is the target area, the precipitation stations should be within the SC_2 sub-catchment.

Line 197-199 ‘The daily … Meteorological Administration [21]’: However, if SC_2 is the target area, the precipitation stations should be within the SC_2 sub-catchment. Why not use grid data such as reanalysis data?

Line 199 ‘Boryeong Meteorological Station’: Where is this station in Fig 1?

Line 208 ‘Figure 5’: There is no mention of Figure 5 in the text. What is 'Actual Evaporation' in the figure? Does it mean that there is observational data?

Line 216 ‘storage data’: Which dam are you talking about?

Line 219 ‘Boryeong Dam catchment’: Isn't the analysis region only for SC_2? Why does this region appear here?

Line 228 ‘soil map’: It would be good to show this soil map in Fig. 2?

Line 229 ‘land cover map’: Does it refer to the same thing as the land use map in Fig. 2?

Line 249 ‘Table 4’: Does Table 4 need to show? If it's necessary, you should at least describe the difference in years and why. What are the units of the values?

Line 260 ‘CAT Model’: When you input daily data, does it output daily data? If I input finer time scale (like hourly), will I get finer time scale output? What are the limits?

Lines 261-271: The details of the CAT model are unclear; please provide a good explanation of why CAT is used and what its features are. Is CAT model the only model for groundwater? What are the advantages of using CAT over other models?

Line 283 ‘Figure 8’: There is no mention of Figure 8 in the text. Is rainfall only in the Impervious zone and PET only in the Pervious zone? Please show the correct diagram.

Lines 283-284, Line 3158-316, and Line 332 ‘(CAT 3.2 user’s manual, 2021)’: Please add it to the reference list.

Lines 319-332: How is the water withdrawal volume calculated?

Lines 350-352: What time scales are you considering for these calculations?

Line 361-363 ‘Figure 11’: There is no mention of Figure 11 in the text. The figure is very blurry. Is it a comparison of daily data? The Figure 11(b) shows a tendency to underestimate when the streamflow is small and to overestimate when the streamflow is large. A discussion of these issues should be included.

Line 369 ‘high accuracy’: Can you really put it that way?

Lines 373-382: Figure 12 shows overestimation of simulated water storage. Instead of showing Figure 12, why not show a that can illustrate the description here?

Line 390 ‘Figure 13’: Where is the water cycle shown?

Line 398 ‘runoff and groundwater recharge increase’: Why are runoff and groundwater recharge increasing?

Line 399 ‘Table 6’: What is the increase compared to? Describe it clearly. The layout should be such that figure and caption are on one page.

Line 454 ‘Table 7’: There is no mention of Table 7 in the text.

Line 462 ‘groundwater elevation’: is it groundwater level?

Line 465-466 ‘WET period than in the DRY period’: Please describe the reason.

Line 468 ‘Figure 16’, Line 475 ‘Tab5le 8’: There is no mention of Table 7 in the text.

Line 475 ‘Table 8’: What is it compared to?

Lines 494-545: The discussion part should be written.

Lines516-517 ‘WET and … SMI drought index’: Where in the text is there a statement related to this?

Reviewer 3 Report

The presented paper is a good example of scientific manuscript and in my opinion, could be recommended to be published. The aim of article is worthy of investigation. Methods, equations, course of analysis are shown clearly and fully. Figures are visible. I am impressed of huge work done. The only one lack of the paper is discussion. Comparing results with other scientists' results would enrich the paper. Based on the comparison, the novelty of the manuscript could be shown clearly. Summarizes, the paper could be published in Water and it is a good example of scientific manuscript.

Round 2

Reviewer 1 Report

The article has been revised very well and can be considered for publication.

Author Response

Dear Reviewer,

I sincerely appreciate your efforts to revise my manuscript properly.

Thank you very much.

Best regards,

Sanghyun Park

Reviewer 2 Report

The edits completed by the authors thus far have improved the clarity of this paper.

I still have a few comments as below.

Line 80 ‘(SMAP)’, Line 89 ‘(SWAT)’: These abbreviations are not necessary, as they are not used in subsequent text.

Line 145 Figure 1 and Line 168 Figure 2: Where did you get the aerial photos shown in Figure 1 and Figure 2? The reference should be clearly indicated.

Line 184 Figure 3: The 'Climate' shown in Figure 3 is not intuitively clear. I don't know what to understand from this figure.

The resolution problem should be solvable; why not just draw directly on the diagram instead of using CAT model system?

You mentioned that the figure 3 shows the boundary of the reservoir catchments, the upstream catchments, and the downstream catchment. Where are these shown? Describe correctly what is being drawn.

Author Response

Dear Reviewer,

I sincerely appreciate your efforts to revise my manuscript properly.

  • The abbreviations in introduction were deleted.
  • The reference of aerial photos in Figure 1 and Figure 2 was added in reference no.25.
  • The climate node in Figure 3 was deleted and the resolution of this figure was improved. I tried to draw this diagram directly, not using CAT system. The boundary lines of each sub-catchments were edited to make it more visible.

Thank you very much.

Best regards,

Sanghyun Park